# Exemplar-based Stylized Gesture Generation from Speech: An Entry to the GENEA Challenge 2022

SAEED GHORBANI, Ubisoft, Canada

YLVA FERSTL, Ubisoft, Canada

MARC-ANDRÉ CARBONNEAU, Ubisoft, Canada

We present our entry to the GENEA Challenge of 2022 on data-driven co-speech gesture generation. Our system is a neural network that generates gesture animation from an input audio file. The motion style generated by the model is extracted from an exemplar motion clip. Style is embedded in a latent space using a variational framework. This architecture allows for generating in styles unseen during training. Moreover, the probabilistic nature of our variational framework furthermore enables the generation of a variety of outputs given the same input, addressing the stochastic nature of gesture motion. The GENEA challenge evaluation showed that our model produces full-body motion with highly competitive levels of human-likeness.

CCS Concepts: • **Computing methodologies → Machine learning**; **Animation**.

Additional Key Words and Phrases: gesture generation, machine learning, style transfer, computer animation

**ACM Reference Format:**

Saeed Ghorbani, Ylva Ferstl, and Marc-André Carbonneau. 2022. Exemplar-based Stylized Gesture Generation from Speech: An Entry to the GENEA Challenge 2022. In *INTERNATIONAL CONFERENCE ON MULTIMODAL INTERACTION (ICMI '22), November 7–11, 2022, Bengaluru, India.* ACM, New York, NY, USA, 9 pages. https://doi.org/10.1145/3536221.3558068

## 1 INTRODUCTION

Non-verbal behavior is an essential part of human interaction, carrying additional information to semantic speech content [8, 19]. Much research has focused on modelling this non-verbal behavior in order to increase virtual agents' life-likeness and appeal, among others, co-speech gesture has received much attention. A key challenge of generating gestures from speech input is the non-deterministic nature of the speech-gesture relationship, where a speech line may be associated with many different motions. Consequently, there is not one true gesture match, but numerous equally appropriate gestures, posing a one-to-many mapping problem. Moreover, each person has their own style of movement, implying that a generative model should capture a wide range motion variation, preserving characters' idiosyncrasies.

Previous works address speaker and gesture variety by modelling and generating gestures for specific speakers individually [2, 7, 18, 27], or by manipulating gesture motion through general statistics such as hand height and velocity [3, 28]. A shortcoming of these approaches is their dependence on the data for creating motion variation, requiring examples of each speaker and style for training the generative model and having trouble generalizing to motion styles beyond the training set.

In this work, we propose style-controllable gesture generation system that can generalize to motion styles beyond training data and alleviates previous limitations to generating gesture styles in a training set-dependent manner. Styles

are encoded from short example motion clips that do not have to be part of the training set. Style representation and conditioning in this manner does not require any explicit labels, circumventing the difficulty of naming gesture styles. Our system is probabilistic, enabling generation of many variations for the same speech input simply by re-sampling, and therefore addressing the stochastic nature of co-speech gesture.

## 2 RELATED WORK

Data-driven approaches to generating co-speech gesture motion aim to learn a mapping from a speech input signal to the corresponding output motion. As this is a one-to-many mapping problem, as discussed above, directly minimizing a difference between target and output motion of a deterministic model can lead to mean pose convergence with small-range motion (e.g. [5] and [14]). Alternatively, the use of adversarial losses have been proposed [6, 7, 21], relying on a second neural network to judge output realism. However, generated motion still fails to match real motion in terms of naturalness and frequently ties with mismatched real motion on appropriateness measures [7, 15].

Employing a probabilistic framework, [3] predict the next pose distribution instead of a fixed pose and by re-sampling from the distribution, a variety of output motion can be generated. For further output variation control, the authors propose style manipulation via four motion parameters, namely gesture speed, height, spacial extent and lateral symmetry. In a similar manner, [28] enable style control via gesture speed, spacial extent, and handedness. [23] instead propose the use of the Laban Effort and Shape parameters to drive gesture style based on the desired personality of a character. Common to all three above approaches is a reliance handcrafted control parameters; these are often too limited to encode a wide range of distinct styles to be learned by the model.

Alternative to hand-crafted style features, a number of works from similar domains have proposed example-based style control. For human locomotion, Aberman et al. [1] perform style transfer via example clips and enable generalization to unseen styles, however, training requires labeled style examples. For dance motion generation, Valle-Pérez et al. [24] present a model conditioned on music and a short style example motion, though the model could not be shown to generalize beyond the motion styles of the training data.

For this work, we looked to recent advances in speech synthesis research. To enable speech stylization by example style generalization beyond the training data, previous works augment a text-to-speech model with a style encoder capturing the general style and prosody of a line, then used to condition the speech synthesis [11, 25, 30]. In addition to their style generalization capabilities, these models produce style latent space that allows for meaningful interpolations. Extending the style generalization, [30] enabled speech synthesis even for unseen speakers.

## 3 SYSTEM OVERVIEW

Our system is composed of three components, (1) a Speech Encoder producing a speech embedding sequence $S$, (2) a Style Encoder summarizing an example motion clip into an embedding vector $\mathbf{e}$, and (3) a Gesture Generator that generates a pose sequence $Y$ from the speech and style embeddings. An overview of our system is shown in Fig. 1.

### 3.1 Speech Encoder

The Speech Encoder, shown in Fig. 2, produces a sequence of speech embedding vectors from the audio input. First, audio features are extracted from the raw audio input as described in Sec. 5. Next, extracted features are passed through 1D convolution layers followed by non-linear operators, followed by a frame-wise linear layer. The kernels of the convolutional layers are set to create a receptive field of about 1 second of speech. The convolutional layers have 64 channels and a kernel size of 3 and 31, respectively, and are followed by a 0.2 dropout rate and ELU activations.

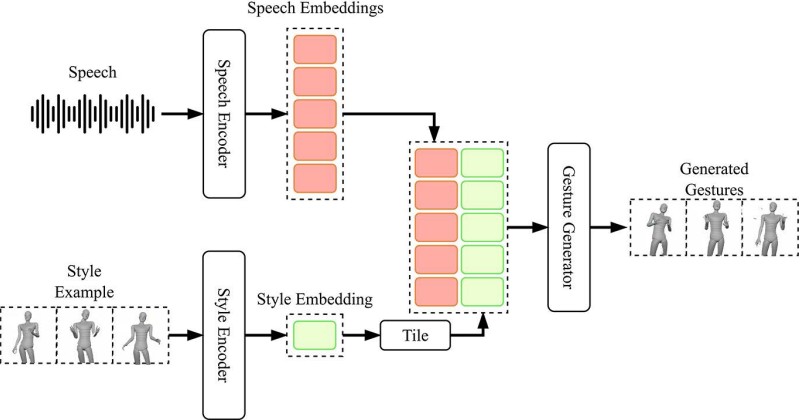

Fig. 1. Overview of our gesture generation system. The Speech Encoder takes the audio input and produces a speech embedding vector. The Style Encoder takes an example motion clip and produces a style embedding vector. The speech and style embedding vectors are fed to the Gesture Generator which generates the output pose sequence.

The Speech Encoder yields a sequence of embedding vectors $S = [\mathbf{s}_0, \mathbf{s}_1, \ldots, \mathbf{s}_{T-1}]$ where $T$ is the number of frames in the sequence and $\mathbf{s} \in \mathbb{R}^{D_S}$. The dimensionality of the speech embedding ($D_S$) is 64.

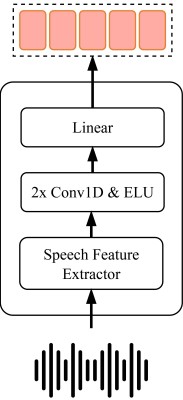

Fig. 2. Architecture of our Speech Encoder.

## 3.2 Style Encoder

The Style Encoder produces a fixed size low dimensional embedding vector representing the motion style of an example clip. Fig. 3a shows an overview of its architecture. This example clip is represented by a sequence of $M$ feature frames, $A = [\mathbf{a}_0, \mathbf{a}_1, \ldots, \mathbf{a}_{M-1}]$. Feature extraction from the example clip is described in Sec. 5. We apply two 1D convolution layers each with a kernel size of 3 and 512 output channels, followed by dropout layers with a drop rate of 0.2. The convolutional layers are followed by a ReLU activation and layer normalization, before the output is passed to a bidirectional GRU of size 512 and a linear projection layer. This yields the parameters $\mu$ and $\sigma$ of the $D_e$-dimensional multivariate Gaussian distribution from which we sample the final style embedding vector $\mathbf{e} \in \mathbb{R}^{D_e}$. The dimensionality

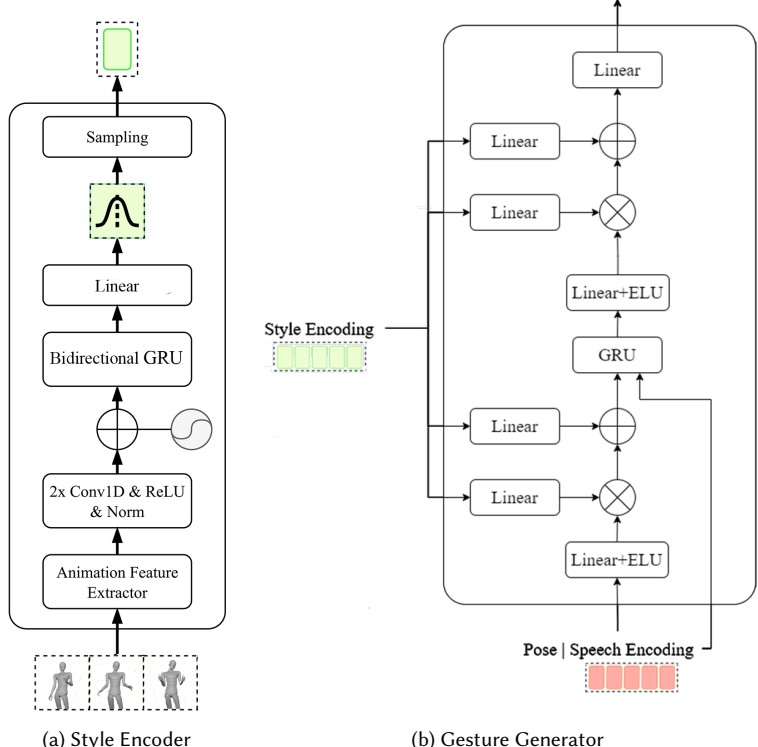

(a) Style Encoder

(b) Gesture Generator

Fig. 3. Architectures of the Style Encoder and the Recurrent Decoder

of the style embedding ($D_e$) is 64. This sampling procedure is described in the Variational Auto-Encoder (VAE) framework [13]. VAEs are known for learning disentangled latent spaces [22] that allow for interpolation [4]. Moreover, they allow us to sample variations at inference time.

### 3.3 Gesture Generator

Our Gesture Generator takes the speech embedding sequence $S$ and the reference style embedding vector $\mathbf{e}$ and produces the final gesture sequence $Y = \{\mathbf{y}_0, \mathbf{y}_1, \dots, \mathbf{y}_{T-1}\}$. The Gesture Generator is depicted in Fig. 3b. The speech embedding is passed through two layers of Gated Recurrent Units (GRU) of size 1024, with the style embedding used as feature-wise linear modulation (FiLM [20]). The Gesture Generator yields the output pose, parameterized similarly to the Style Encoder as $\mathbf{y} = [\boldsymbol{\rho}_p, \boldsymbol{\rho}_r, \dot{\boldsymbol{\rho}}_p, \dot{\boldsymbol{\rho}}_r, \mathbf{r}_r, \dot{\mathbf{r}}_p, \dot{\mathbf{r}}_r]$, where $\boldsymbol{\rho}_p, \boldsymbol{\rho}_r, \dot{\boldsymbol{\rho}}_p, \dot{\boldsymbol{\rho}}_r$ are joint local translations and rotations along with their velocities, and $\dot{\mathbf{r}}_p$ and $\dot{\mathbf{r}}_r$ are the character root translational and rotational velocity local to the character root transform. $\mathbf{r}_p \in \mathbb{R}^3$ and $\mathbf{r}_r \in \mathbb{R}^4$ are the position and orientation of the root (represented as quaternions), respectively, which are updated using the root translational and rotational velocities at each frame.

### 4 TRAINING & LOSSES

Our network is trained end-to-end with the Rectified Adam optimizer [17] with a learning rate of $10^{-4}$ and a decay factor of 0.995 applied at every 1k iterations. We use a batch size of 32 sequences of length $T = 128$ frames (4.26 seconds)

and train the network for 100k iterations based on visual result quality. The conditioning style example used for training encompasses the 128 target frames but is extended by surrounding frames from the source animation clip. The number of surrounding frames is chosen randomly to create sequence of length $M$ spanning from 128 to 256. This ensures de-correlation between styles and clip length. The model is trained using its own predictions (i.e. without teacher forcing), letting the model learn to recover from its own errors and therefore producing more robust results.

Framing our model as a conditional VAE, the objective is to maximize the evidence lower bound (ELBO) of the marginal log likelihood of gesture motion given a speech sequence. The training loss can then be expressed as:

$$\mathcal{L} = \mathbb{E}_{q(\mathbf{z}|\mathbf{e})} \left[ -\log p \left( Y \mid S, \mathbf{z} \right) \right] + D_{KL} \left( q \left( \mathbf{z} \mid \mathbf{e} \right) \| p \left( \mathbf{z} \right) \right)$$
$$= \mathcal{L}_{recon} + D_{KL} \left( q \left( \mathbf{z} \mid \mathbf{e} \right) \| p \left( \mathbf{z} \right) \right) \tag{1}$$

The first term represents the expected negative log-likelihood of the gesture motion, i.e. the reconstruction loss. The second term is the regularization term in the form of the Kullback–Leibler divergence between the posterior distribution $q \left( \mathbf{z} \mid \mathbf{e} \right)$ predicted by the style encoder and the prior distribution $p \left( \mathbf{z} \right)$, which is a standard multivariate Gaussian distribution. For weighting the regularization term, we use cost annealing as proposed by [4].

Our reconstruction loss divides into the following terms:

$$\mathcal{L}_{recon} = \lambda_p \mathcal{L}_p + \lambda_r \mathcal{L}_r + \lambda_{vp} \mathcal{L}_{vp} + \lambda_{vr} \mathcal{L}_{vr} +$$
$$\lambda_{dp} \mathcal{L}_{dp} + \lambda_{dr} \mathcal{L}_{dr} + \lambda_f \mathcal{L}_f \tag{2}$$

where $\mathcal{L}_p$, $\mathcal{L}_r$, $\mathcal{L}_{vp}$, and $\mathcal{L}_{vr}$, are the mean absolute error (MAE) between predicted and target joint positions, rotations, translational velocities, and rotational velocities, respectively, in both local and world spaces. In addition to direct velocity predictions, $\mathcal{L}_{dp}$ and $\mathcal{L}_{dr}$ penalize the velocity MAE by computing the translational and rotational velocities in the local and world spaces on-the-fly via finite-difference. We took inspiration from the reconstruction loss for Learned Motion Matching proposed by [10]. Lastly, $\mathcal{L}_f$ penalizes the MAE for the facing direction in the world space to prevent any character rotational drift. Loss terms are empirically weighted to be on a similar scale.

## 5  DATA PREPARATION

The GENEA challenge provided full-body motion files with synchronized audio and transcriptions from the *Talking With Hands 16.2M* conversational dataset [16]. The data contains long periods of silence, low volume audio, cross-speaker audio bleeding, speaker data with little to no hand gesture motion and noisy finger motion. All of which could hinder model training. To address this, we filter and process the data as follows:

We remove long periods of silence (> 0.5 seconds) in the recordings based on the speech transcripts. This process also removes longer sections in which an interlocutor is speaking causing audio bleeding into the recording. We normalize the loudness of all recordings to -20 dB LUFS [12] to address low and varying volume audio sections.

Regarding animation data, we identify and remove the samples with low average hand velocity, which would result in a static generated gestures. We compute the average hand velocity in the root coordinate and remove all clip with less than $10cm/sec$. In addition, we remove finger motion data due to the poor finger motion quality which would lead to unnatural looking hands.

We extract low-level features from the raw audio data. We extract spectrograms using an FFT Hanning-window of 50 ms and a hop length of 12.5 ms. We project the spectrogram into the mel frequency scale, and use the log amplitude of each of the 80 channels. We additionally extract the total energy per frame as a secondary feature. Finally we re-sample the speech features to 30 frames per second to match the frame rate of the provided motion files .

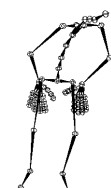 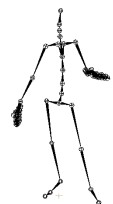 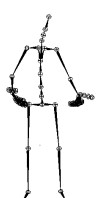 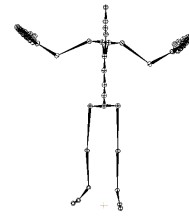

Fig. 4. Samples of generated gestures given different style examples. From left to right: Tired, Relaxed, Sad, Oration. These samples are generated by training our model with a dataset covering a variety of different styles. The style examples are then provided to the model from a held-out validation data. The oration style was not part of the training data.

For motion data, we represent each frame by a feature vector $\mathbf{a} = [\boldsymbol{\rho}_p, \boldsymbol{\rho}_r, \dot{\boldsymbol{\rho}}_p, \dot{\boldsymbol{\rho}}_r, \dot{\mathbf{r}}_p, \dot{\mathbf{r}}_r]$ where $\boldsymbol{\rho}_p \in \mathbb{R}^{3j}, \boldsymbol{\rho}_r \in \mathbb{R}^{6j}$ are the joint local translations and rotations, $\dot{\boldsymbol{\rho}}_p \in \mathbb{R}^{3j}$ and $\dot{\boldsymbol{\rho}}_r \in \mathbb{R}^{3j}$, are the joint local translational and rotational velocities, and $\dot{\mathbf{r}}_p \in \mathbb{R}^3$ and $\dot{\mathbf{r}}_r \in \mathbb{R}^3$ are the character root translational and rotational velocity local to the character root transform. $j$ corresponds to the number of joints. Joint rotations are represented by 2-axis rotation matrix and joint and root rotational velocities are specified using the scaled angle axis representation as in [31]. The root rotation is obtained by projecting the *z-axis* of the hip joint onto the ground. The root position of the character is computed by projecting the position of the second spine joint on the ground. We represent the root position with one of the spine joints because there is less high-frequency jitters when compared to the hip joint, which leads to a more stable character position.

## 6 RESULTS

To produce the animated gestures sent for the challenge evaluation, we chose an animation sample from the validation set for each speaker in the test set. We encoded the style examples and used the embeddings to generate gesture for the test speech files. We used the same data preparation procedure described in Section 5 for the validation and test data. We provided the same generated samples for both full-body and upper-body evaluations.

Our model produced highly competitive results in the evaluation of the GENEA Challenge 2022 [29]. For human-likeness, the full-body motion produced by our model ranked second, placing only behind ground truth and the entry ranking on-par with ground truth (condition FSC in Table 1 and Figure 2 of [29]). Although we trained our model only on the full-body data with global movement modelling and more complexity, our model was ranked third in the upper-body evaluation (Condition USO in Table 1 and Figure 2 of [29]). Regarding appropriateness for speech, our system performed similarly to most other entries, with all entries falling significantly behind ground truth and obtaining similar scores. (refer to Figure 5 in GENEA challenge main paper [29]). The GENEA challenge also included a number of objective, numeric measures of performance, namely average joint jerk, positional speed profile, and canonical correlation. However, scores on these measures did not correlate well with perceptual results and therefore do not appear to be informative of output quality.

We visually validated that our model generalizes to varied styles. We used a proprietary dataset that depicts a large number of very different styles. Fig. 4 shows some skeleton poses from generated motion using conditioning examples from different styles. The conditioning clips were not part of the training data, but other examples from the same style were in the dataset. However, we also tested with the *oration* style that was not seen during training to explore the zero-shot capability of our model. Even without having seen similar oration examples before, the model reproduced the pose and stance of the orator with raised hands that move to mark speech inflections. We only briefly discuss the

generalization capabilities of the model due to space limitation and because zero-shot gesture generation is not part of the challenge.

## 7 DISCUSSION

Our system performed overall well in the challenge and showed highly competitive performance. Our method was primarily designed for zero-shot style control by example, meaning that our model can generate gestures for unseen styles, without retraining on the new styles. There is a trade-off between increased generalization capabilities and tailoring a model for a specific task. Nonetheless our system performed overall well and showed highly competitive performance. Moreover, our model is probabilistic, allowing generation of many output variations given the same input. While not evaluated in the challenge, we believe that strong generalization capabilities and being able to generate diverse variations for the same audio clip are two highly desirable qualities for a generator model.

While ranking second and third in the human-likeness subjective evaluations, this performance was not reflected in the objective evaluations. However, by comparing the overall challenge results, it appears that these measures did not strongly correlate with perceptual evaluations across all methods. This led the organizers to argue that objective evaluation is currently not particularly meaningful for assessing gesture generation quality, and that subjective evaluations are the best option for such assessment. This is in line with the conclusions of [26]. We believe that distance metric, in their current form, will always fail to accurately capture perceptual gesture quality; as distance is measured with respect to a ground truth sequence, it is implicitly assumed that anything that deviates from it is of lesser quality, however, several gesture sequences may be equally appropriate for the same speech clip, but only those closer to a particular ground truth realization will be correctly scored.

Based on our experience participating in the challenge, we highlight the importance of data preparation. We observed significant improvement in our model performance after applying the data preparation operations described in Sec. 5.

In future work, we would like to iterate upon our data preparation methods to further improve result quality. It is likely that some non-informative segments remain despite our filtering operation. Moreover, some data augmentation techniques could be applied to audio files as well as animation clips (e.g. mirroring, speed manipulations). Given that numeric measures do not strongly correlate with perceptual quality we will investigate the effect of adversarial losses on subjective results. For instance, as done in traditional generative adversarial networks [9], a second network trained to distinguish between generated and ground truth animation could potentially guide our system in generating more realistic animations. We could also implement cycle consistency loss similarly to [32] to ensure that generated gesture accurately reproduce the style of the conditioning example.

## 8 CONCLUSION

We described our entry to the GENEA 2022 Gesture Generation Challenge. Our system allows for exemplar-based style-control via short motion clips. It operates in limited data settings and alleviates the difficulty of devising a fine-grained taxonomy for gesture styles. In addition, our system can generate many output variations for the same input speech segment. The challenge results showed that our method produces competitive natural-looking motions when compared competing entries, while offering the aforementioned capabilities. Perhaps unsurprisingly, we found that data preparation is key in obtaining good performance, and that since current objective measures do not strongly correlate with subjective evaluations, losses used during training should be revisited to better align with human perception. Finally, we plan on conducting full evaluation of the zero-shot learning capabilities of our model and assess disentanglement and control over our learned style embedding space.

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
