# OpenReview forum: "Exemplar-based Stylized Gesture Generation from Speech: An Entry to the GENEA Challenge 2022"
_ACM.org/ICMI/2022/Workshop/GENEA — GENEA Challenge & Workshop 2022 Mainproceeding_

### Official Review · Reviewer_G94b · 2022-08-07
**Gesture generation system with gesture style control through short examples.**

**Rating:** 8
**Confidence:** 4

**Review:**

Quality:
Most of details about the methods were explained, but diagrams with evaluation results and their discussion is missing. Also some details for training and generating samples for evaluation are missing, in specific information on how the zero-shot examples are used.

Clarity:
Paper is well written.

Originality:
Transferred recent developments of using examples to control speech styles from speech synthesis to gesture synthesis.

Significance:
Full-body model ranked second for human-likeness.

Pros:
- Method obtained very good results for human-likeness evaluation.

Cons:
- Result section lacks evaluation results and detailed discussion of the outcomes.
- Some normalization details on data preparation are missing.
- It is not clear how the different examples are used in the style encoder during training.
- It is not clear how different examples were used to generate the motion for evaluation.


Suggestions for improvement:
- More emphasis on contributions, in specific model characteristics which are different from SOTA (abstract and related work).
- Given that authors saw significant improvement after applying data preparation, it would be good to include more details on that step. What were the parameters for audio normalization? What was the threshold used to distinguish low average hand movements?
- Include evaluation diagrams and discuss results in more detail.
- Describe how different examples are used in the style encoder during training. Is one example used to train with all data? Is a different example used for each data sample?
- Describe how different examples were used to generate the motion for evaluation. For each generated sample a different example?
- Some typos: Line 9 "embedding. The", line 193 "This yields the parameters...", line 216 add superscript for e^-4, line 329 "alleviates", line 330 "an fine-grained".

---

### Official Review · Reviewer_E7jV · 2022-08-08
**A generally well written paper with an interesting approach beneficial to the gesture generation community. Style control is emphasised, but lacks examples in the text. The method is limited by not predicting hand motion and some details are missing for reproducibility.**

**Rating:** 5
**Confidence:** 4

**Review:**

### Overview

The authors introduce a new method to generate stylised gestures using a zero-shot learning approach to create a style embedding. A variational framework allows many different animations to be produced from a single example of audio and a motion sequence for style transfer. Some results are discussed but readers are also referred to the main GENEA paper.

The zero-shot learning of styles is an interesting approach providing scientific value and performed well in the challenge. The “System Overview” and “Training & Losses” sections are fairly detailed and well presented. There is a good discussion on the benefits of data processing. The figures are clear and well presented. However, I have some questions that could be addressed in the text. Some extra detail and figures could be useful to reinforce some claims as detailed below:

### Comments and Suggestions

1. There is a lot of emphasis on style control, however, there are no examples provided. For example, could a figure of sequences using the same audio input, but different style conditions be included? e.g. Conditioned from the same speaker identity but two different sequence samples and a style sequence from a different speaker identity.
2. There are claims the model can generate gestures for unseen speaker styles. There are currently no results or examples shown. While these are not tested in the challenge evaluation itself, could further results/discussion be included regarding the style transfer? Are unseen speaker styles well described by the embedding or is there a strong bias to training speaker styles? Were any speaker identities held out from the training data to strengthen this claim?
3. The style encoding is generated from “one sample of each speaker” (L221). Could more information on the sampling process be provided for reproducibility? For example, is this sampled from the training motion data? At test inference time, was a different sample used for each sequence?
4. L261 The removal of hand/finger motion is a major limitation of the system in regards to the challenge. Could a discussion/justification of this design choice be included?
5. L247 “Loss terms are empirically weighted” These weight values would be good to include for reproducibility.
6. In the “Data Preparation” section, it is noted that the audio and animation data are normalised. It is unclear what method of normalisation is performed here and may limit reproducibility. Could these methods be included?
7. Participant IDs are not explicitly stated in the results for cross-referencing the main GENEA paper. These should be included in the text.
8. L301, “While ranking **second and third** in the human-likeness subjective evaluations” Only second in the full-body tier was mentioned in the results section. Was this method submitted to both the full-body and upper-body tiers? Could the results be updated to reflect this along with the participant IDs?
9. L274 Describes deriving root position and rotation from the second spine and hip joints respectively. Two different joints were chosen to represent the position/rotation of the root instead of a single joint. It is unclear why this method was used rather than the root joint provided in the dataset. Could justification be provided in the text?
10. L261 “low average hand velocity to remove data” to ensure reproducibility, could the threshold used to describe low average hand velocity be included here and preferably how that threshold was derived?
11. L206 missing the citation for FiLM. [1]

[1] Perez, Ethan, et al. "Film: Visual reasoning with a general conditioning layer." *Proceedings of the AAAI Conference on Artificial Intelligence*. Vol. 32. No. 1. 2018.

---

### Decision · Program_Chairs · 2022-08-11

**Decision:**

Accept (Main proceeding)

**Comment:**

Congratulations, your paper was accepted to the GENEA Challenge 2022 and is going to be published in the main ICMI Proceedings, given that the camera-ready version is provided on time.

Reviewers were generally happy with the paper but had a number of suggestions for improvement. Those mainly relate to clarifications and reproducibility enhancement. See below for the full reviews.

We suggest that the authors carefully consider the feedback received from the reviewers and use it to improve their manuscript for the workshop camera-ready submission on August 17th.